# The Economic and Occupational Impact of Mental Health-Related Temporary Work Disabilities in Spanish Workers During and After the COVID-19 Pandemic: A Longitudinal Study

**DOI:** 10.3390/healthcare13060618

**Published:** 2025-03-13

**Authors:** Eva María Gutiérrez Naharro, José Antonio Ponce Blandón, Amalia Sillero Sillero, José Fernández Sáez

**Affiliations:** 1Escoles Universitàries Gimbernat, Adscrita a Universitat Autònoma de Barcelona, 08174 Sant Cugat, Spain; eva.gutierrez@eug.es; 2Nursing Department, Faculty of Nursing, Physiotherapy and Podiatry, University of Seville, 41009 Seville, Spain; japonce@us.es; 3Facultad de Enfermería, Campus Terres de l’Ebre, Universitat Rovira i Virgili, 43500 Tarragona, Spain; jfernandez@idiapjgol.info; 4Servei Atenció Primaria Terres de l’Ebre, Institut Català de la Salut, Unitat de Suport a la Recerca Terres de l’Ebre, Fundació Institut, Universitari per a la Recerca a l’Atenció Primaria de Salut Jordi Gol i Gurina, 43500 Tarragona, Spain

**Keywords:** mental health, COVID-19, temporary work disability, occupational health, anxiety disorders

## Abstract

Background: The COVID-19 pandemic has affected mental health worldwide, exposing gaps in managing work-related disabilities. In Spain, Mutual Collaborators with Social Security play a pivotal role in managing temporary work disabilities associated with mental health conditions. Objectives: This study aimed to describe and analyze the prevalence, characteristics, and economic burden of mental health-related temporary work disabilities in Spain during the COVID-19 pandemic and recovery (2020–2022). Methods: A prospective longitudinal design was used, drawing on data from CHAMAN, a secure and nationally representative database managed by Asepeyo Mutual Insurance and maintained by Mutual Collaborators with Social Security (MCSSs). The study included workers who experienced temporary work disabilities due to mental health disorders diagnosed according to the International Classification of Diseases, Tenth Revision, during the COVID-19 state of alarm (14 March–21 June 2020) and in the following two years (2021–2022). The key variables were demographics, absence duration, cost per case, and relapse rates. The analyses included descriptive statistics and the Mann–Whitney U, Chi-square, and logistic regression tests. Results: In 2020, 5135 cases were recorded, with an aggregate cost of approximately EUR 44.8 million. Regional analysis showed that Catalonia, Castile–La Mancha, and Castile–León accounted for over half the expenditure, whereas costs were lower in other regions. Marked declines in incidence and costs were observed in 2021–2022, suggesting adaptation to post-pandemic conditions. However, significant gender disparities persisted, with women experiencing higher relapse rates and prolonged absences. Generalized Anxiety Disorder and Major Depressive Disorder were the most common diagnoses, particularly in sectors such as retail, elderly care, and call centers. Conclusions: Although the incidence and cost of temporary work disabilities declined, persistent gender inequalities, regional disparities, and sector-specific risks highlight the need for targeted mental health interventions. Enhancing psychosocial support, adopting gender-sensitive workplace policies, and improving regional healthcare infrastructure are essential to promote workforce well-being and ensure economic sustainability.

## 1. Introduction

The COVID-19 pandemic has profoundly highlighted the interconnectedness be-tween health systems, domestic economies, and governance. Government decisions have shaped healthcare infrastructures, regulations, and guidelines and determined access to medication, health coverage, and resource allocation, influencing both immediate public health outcomes and long-term societal resilience [1,2]. In particular, the pandemic ex-posed the fragility of health and social security systems in managing large-scale crises, revealing gaps in how mental health issues translate into economic and occupational challenges [3].

COVID-19 responses globally saw health policies extend beyond the traditional remit of Ministries of Health, involving multidisciplinary expertise to safeguard public health and economic stability [4]. Countries with experience in managing health-related disasters like Ebola adopted whole-of-government strategies, translating evidence-based research into effective policies [5]. However, while biomedical perspectives dominated decision-making bodies, broader occupational health and socio-economic dimensions were often overlooked [6].

The mental health impact of COVID-19 was not limited to the direct effects of the virus but also stemmed from stringent public health measures such as lockdowns, social isolation, and economic uncertainty [7,8]. In Spain, the declaration of a state of alarm on 14 March 2020 imposed strict confinement measures that significantly affected the psychological well-being of the population [9]. Prolonged lockdowns have been associated with adverse mental health outcomes, including acute stress, anxiety disorders, depression, and post-traumatic stress symptoms [10].

Beyond these immediate effects, the prolonged disruption of daily life, financial instability, and job insecurity contributed to increased psychological distress across diverse occupational sectors. The fear of job loss, combined with unprecedented workloads for essential workers and the challenges of remote work for others, led to widespread burnout and emotional exhaustion [10]. Additionally, the pandemic disproportionately affected certain demographics, such as women, younger workers, and individuals in precarious employment, exacerbating pre-existing inequalities in mental health outcomes [11]. Studies suggest that chronic stress and anxiety linked to job instability may have long-term consequences, including an increased risk of relapse in individuals with pre-existing mental health conditions [12].

Despite extensive research on COVID-19’s clinical manifestations and therapeutic interventions, there remains a significant gap regarding the pandemic’s occupational consequences, particularly concerning mental health-related temporary work disability (TWTD) and its economic burden [13].

While existing studies primarily focus on healthcare workers, educators, and vulnerable groups, comprehensive data on the broader workforce are lacking [14,15]. Moreover, the long-term effects of mental health conditions on work capacity, including relapse rates and economic sustainability, are often neglected [16,17].

Mutual Collaborators with Social Security (MCSSs) played a pivotal role in managing work-related absences, particularly mental health. In Spain’s National Social Security System, MCSSs are non-profit organizations authorized to manage public social security services, including work-related accidents, occupational diseases, and temporary work disabilities arising from common contingencies [18,19]. They bridged the gap between healthcare services and the workforce, ensuring medical care and administering financial support during work incapacity.

Several critical concepts are key to understanding this study. “Common contingency” refers to non-occupational illnesses or accidents that result in temporary work disability, managed through the public health system but administratively overseen by MCSSs [20]. “Temporary work disability” denotes the medically certified inability to perform work duties due to health conditions [21], while “work absenteeism” encompasses any work absence increasingly linked to mental health disorders in the post-pandemic context [22].

While prior studies have examined the psychosocial impact of COVID-19 on workforce well-being, most studies have employed short-term or cross-sectional designs and focused on specific groups (e.g., healthcare workers). Longitudinal analyses investigating both the economic and occupational consequences of mental health-related temporary work disability (TWTD) remain scarce, particularly over a multi-year period. Against this backdrop, this study aims to fill this knowledge gap by analyzing TWTD trends, costs, and relapse rates among Spanish workers during and after the pandemic. In addition to assessing the prevalence and characteristics of TWTD, this research highlights the substantial economic burden associated with these absences, providing essential insights for workplace policies, mental health interventions, and social support systems. Furthermore, by examining the role of Mutual Collaborators with Social Security (MCSSs) in managing mental health-related TWTD, this study offers a longitudinal perspective on the intersection of public health crises, occupational health, and economic sustainability, contributing to the broader understanding of how large-scale disruptions affect the workforce.

## 2. Materials and Methods

### 2.1. Design

This study adopted a longitudinal prospective design, following the same subjects over time to analyze changes and trends in mental health-related temporary work disability (TWTD), its economic burden, and relapse rates. The research encompassed three phases: an initial assessment in 2020, evaluating the prevalence and economic cost of TWTD during the COVID-19 lockdown; a follow-up period from 2021 to 2022, tracking evolution, relapse rates, and sustained economic impact; and a comparative analysis, examining demographic, occupational, and regional variations over time.

### 2.2. Study Population and Sample

The target population consisted of all workers in Spain affiliated with Asepeyo who experienced temporary work disability due to mental health conditions during the COVID-19 pandemic and the subsequent two years. Workers were included if they had a temporary work disability due to commoncontingency and were diagnosed with a mental health disorder. The eligibility criteria required that the disability had been initiated during the official COVID-19 state of alarm (14 March 2020–21 June 2020) or in the following two years (2021–2022). All cases had to be managed by Mutual Collaborators with Social Security, with diagnoses coded according to the International Classification of Diseases, Tenth Revision (ICD-10) for mental health conditions. The study initially considered 6082 workers. To ensure data integrity and focus on COVID-19-related TWTD cases, workers were excluded if they had a pre-existing TWTD before the COVID-19 lockdown (*n* = 54), had insufficient data to be included in the study (*n* = 108), or did not attend follow-up assessments (either telephone or in-person) (*n* = 785). After applying these exclusion criteria, the final sample consisted of 5135 workers, followed over two years (2021–2022) to analyze new TWTD cases, relapses of mental health-related TWTD, economic burden, and workdays lost.

A flowchart (Figure 1) illustrates the step-by-step sample selection process and ensures transparency in cohort selection. The CHAMAN database, which consolidates data from all Autonomous Communities (ACs), ensured that the sample was representative of the national working population affiliated with Asepeyo. The strict inclusion and exclusion criteria enhanced the reliability and homogeneity of the data, minimizing variability from non-standardized evaluations.

### 2.3. Context and Follow-Up Years

The study covered all Autonomous Communities (Acs). Data collection spanned three years (2020–2022), ensuring a comprehensive evaluation of TWTD cases, relapses, and economic burdens across regions. This extended follow-up period is particularly significant as it captures both the acute phase of the COVID-19 pandemic and the subsequent recovery phase.

### 2.4. Data Collection

Data were retrieved from the CHAMAN (Center for Histories and Analysis of Mutual Insurance in Accidents and Notifications) database, an authorized registry maintained by MCSSs. Patient records were pseudo-anonymized and coded using ICD-10 for mental health disorders. Key variables extracted include demographics (age, sex, region), TWTD duration (days of absence), sector of employment (CNAE classification), economic impact (cost per TWTD case by region and gender), and relapse rates (subsequent TWTD occurrences in 2021–2022). These data were recorded as part of MCSSs’ routine operational processes and updated continuously from 2020 onwards. The research analysis protocol was finalized and received ethical approval in 2022, covering the retrospective use of earlier data and prospective inclusion of new cases throughout the follow-up period.

### 2.5. Statistical Analysis

A combination of descriptive and inferential statistics was used. The Mann–Whitney U test was applied to compare economic burden across groups, while the Chi-square test was employed to assess sector-based TWTD prevalence. Logistic regression models were used to identify risk factors for TWTD relapse. In addition to traditional statistical significance testing, effect size measures were used to assess the magnitude of the observed differences. Cohen’s d was applied for pairwise comparisons of economic burden and TWTD duration between groups. Statistical analyses were performed using SPSS version 26.0 (IBM, Armonk, NY, USA). A significance threshold was set at *p* < 0.05 to ensure statistical reliability.

### 2.6. Ethical Considerations

Although this study followed a longitudinal prospective approach from 2020 to 2022, the dataset was initially gathered through routine administrative practices by Mutual Collaborators with Social Security (MCSSs). The research team formalized its analysis plan and obtained ethical approval in 2022 (Approval Code: 2022/70-MLA-ASEPEYO; Approval Date: 6 October 2022), which covered the retrospective use of data collected since 2020 and the prospective inclusion of cases recorded throughout 2022. All records were anonymized or pseudo-anonymized prior to analysis, and no personal identifiers were accessed by the investigators. All procedures adhered to the ethical principles of the Declaration of Helsinki (October 2013, Fortaleza, Brazil).

## 3. Results

### 3.1. Economic Burden of TWTD Across Regions

The analysis of mental health-related temporary work disabilities (TWTD) during and after the COVID-19 lockdown highlights significant variations across Spain in terms of economic burden, relapse trends, gender disparities, and sectoral impact. In 2020, during the state of alarm, a total of 5135 TWTD cases were recorded due to mental health disorders, leading to a total economic cost of EUR 44,839,628.03. The financial impact varied substantially across regions, with Catalonia bearing the highest cost (EUR 14.46 million, 32.3% of the total), followed by Castile–La Mancha (EUR 6.78 million, 15.1%) and Castile–León (EUR 4.85 million, 10.8%), collectively accounting for over half of the total expenditure. In contrast, the lowest economic burden was observed in the Balearic Islands (0.5%), Asturias (0.7%), and the Canary Islands (0.8%) (Table 1)

### 3.2. Gender Differences in TWTD Economic Costs

The financial impact of TWTD cases in 2020 varied significantly across Autonomous Communities (CCAA). Cantabria recorded the highest average economic cost per TWTD case (EUR 3390.50), followed by Madrid (EUR 2986.75). Conversely, the lowest economic burdens were observed in Extremadura (EUR 1538.43) and La Rioja (EUR 1751.67). When analyzing gender differences, men generally incurred higher financial costs per TWTD case compared to women in most regions. However, exceptions were noted in the Valencian Community and Galicia, where women had slightly higher average TWTD-related costs than men. Significant gender differences in TWTD-related costs were found in Catalonia and the Basque Country (Table 2).

### 3.3. Temporal Trends in TWTD Cases

Between 2020 and 2022, the number of mental health-related temporary work disability (TWTD) cases decreased significantly, indicating a reduction in work absences due to psychological conditions following the peak of the COVID-19 pandemic. In 2021, 308 TWTD cases were recorded, which declined to 260 cases in 2022, reflecting a progressive recovery in workforce stability.

Gender disparities remained consistent, with women exhibiting higher relapse rates than men. Women accounted for 60% of TWTD cases in 2021 and 62% in 2022, whereas men represented 40% and 38%, respectively. The proportion of TWTD cases directly linked to COVID-19-related psychological distress also declined. In 2021, 6% of cases in women and 2% in men were attributed to COVID-19, while in 2022, these figures dropped to 1% and 2%, respectively. This trend suggests gradual adaptation to post-pandemic work conditions (Table 3).

### 3.4. Duration of TWTD Cases and Gender Differences

The average duration of TWTD cases varied across gender, with women accumulating more workdays lost than men throughout the study period. In 2021, the total number of workdays lost due to TWTD was 7866 days for women and 4945 days for men. By 2022, these figures had declined to 6425 days for women and 3685 days for men, indicating a progressive reduction in absenteeism and suggesting potential improvements in workplace mental health support (Figure 2).

### 3.5. Economic Burden of TWTD

The economic impact of TWTD cases followed a similar downward trajectory, with a significant reduction in financial costs over time. In 2021, the total economic burden associated with TWTD reached EUR 13,373,075.35, of which EUR 5,611,882.96 corresponded to men and EUR 7,761,192.39 to women. By 2022, the total cost had decreased to EUR 1,089,587.52, with men accounting for EUR 621,598.08 and women for EUR 467,989.44. The decline in workdays lost and overall TWTD cases contributed to this reduction in economic burden, although the gender gap in financial impact remained evident (Table 4).

### 3.6. Evolution of Mental Health Diagnoses from 2020 to 2022

The prevalence of specific mental health disorders contributing to TWTD evolved. Generalized Anxiety Disorder (GAD) remained the most frequently reported diagnosis, though its incidence declined from 22% in 2021 to 13.42% in 2022. Other common diagnoses, including Major Depressive Disorder (MDD) and Adjustment Disorder with Behavioral Disturbance, remained significant contributors to TWTD cases, with relatively stable prevalence across the study period.

Conversely, specific disorders, such as Acute Stress Reaction and Panic Disorder, exhibited a sharp decline, likely reflecting reduced uncertainty, improved resilience, and better workplace adaptation post-pandemic. Despite the overall reduction in TWTD cases, gender disparities persisted, with women consistently exhibiting higher relapse rates than men (Figure 3).

### 3.7. Occupational Sectors Most Affected by TWTD

Specific occupational sectors were disproportionately impacted by mental health-related TWTD, particularly those characterized by high levels of social interaction and emotional stress. The retail sector, specifically textiles, clothing, footwear, and cosmetics, reported the highest number of TWTD cases (490). Other high-risk sectors included residential care for the elderly (343 cases), non-specialized retail trades (238 cases), and call center activities (189 cases).

Additional sectors with a significant TWTD burden included the pharmaceutical industry (172 cases), healthcare and hospital activities (159 cases), general education (152 cases), public administration (141 cases), and general building cleaning services (118 cases).

The most prevalent diagnosis across these sectors was Generalized Anxiety Disorder, particularly in high-pressure environments that involved rapid operational changes and frontline responsibilities. Work environments such as call centers, travel agencies, private security services, and healthcare institutions demonstrated higher levels of anxiety-related TWTD cases, reinforcing the association between occupational stress and mental health-related work absences (Figure 4).

## 4. Discussion

The present longitudinal study provides a comprehensive overview of mental health-related temporary work disabilities (TWTDs) in Spain from 2020 to 2022, illustrating a sharp increase during the initial COVID-19 lockdown and a notable reduction in the subsequent years. These trends mirror observations from previous large-scale health emergencies, such as SARS and MERS, in which heightened psychological distress stemming from social isolation and economic instability led to elevated rates of anxiety and depressive disorders [23,24]. By late 2021 and into 2022, the gradual easing of restrictions, enhanced psychosocial resilience, and implementation of targeted workplace mental health programs may have collectively mitigated the incidence of TWTD. Similar declines have been observed in other epidemic contexts once public health measures stabilized and individuals adapted to changed social and occupational norms [25,26].

Despite this overall decrease, the economic and social burden of TWTD remains substantial. The aggregated costs, peaking at EUR 44.8 million in 2020, declined to EUR 13.3 million in 2021 and finally dropped to EUR 1.08 million in 2022. These figures align with broader research linking mental health-related absences to significant productivity losses and national healthcare expenditures [27]. Nonetheless, striking regional disparities persist, with Catalonia, Castile–La Mancha, and Castile–León incurring the highest expenditures, while other communities like the Balearic Islands and Asturias consistently reported lower burdens. This uneven distribution of costs points to regional imbalances in industrial composition, mental healthcare infrastructure, and resource allocation. One potential solution is to develop regionally tailored mental health networks that balance economic capacities with clinical demand. Specifically, policy-makers could channel targeted investments toward regions with elevated TWTD rates, ensuring adequate mental health staffing and telehealth services [28].

A conspicuous finding is the difference in TWTD patterns between men and women. While men generally incur higher average costs per TWTD case—possibly due to wage differentials and occupational roles—women consistently exhibit higher relapse rates and an overall more significant share of cases. This phenomenon resonates with the literature on gender-specific stressors such as caregiving responsibilities and precarious employment [29,30]. One way to address these gender disparities is for employers and social security authorities to integrate flexible work policies, provide subsidized or on-site childcare, and sponsor regular psychosocial evaluations. These measures could alleviate the heightened emotional labor and stress often shouldered by women in service-oriented or frontline positions. Additionally, the robust enforcement of pay-equity laws and programs aimed at career advancement for women might reduce the long-term financial strain of TWTD [31,32].

Specific occupational sectors stand out for disproportionately high TWTD incidence, including retail, residential elderly care, and call centers. These roles involve extensive public interaction and emotional labor demands, exacerbated during the pandemic by staffing shortages, fear of contagion, and shifting protocols [33]. Even healthcare workers—who might otherwise be assumed to possess greater mental health literacy—reported substantial TWTD, highlighting the intense pressures placed on frontline professionals [24,34]. In response, workplaces could implement stress-reduction initiatives, such as resilience training, the rotation of high-stress assignments, and structured peer support groups, complemented by organizational-level strategies like adjusting workloads and staffing. Evidence from prior interventions suggests that early detection and targeted psychosocial support effectively reduce the severity and duration of absences [35].

Although the considerable decline in TWTD by 2022 is encouraging, the persistent prominence of Generalized Anxiety Disorder (GAD) among these cases indicates that pandemic-related stressors—and their associated occupational tensions—have not been fully resolved [36]. In many scenarios, job insecurity and disruptions in work–life balance extend beyond the acute crisis, increasing the risk of recurrent or lingering mental health problems. A potential solution is for Mutual Collaborators with Social Security (MCSSs) and employers to adopt proactive approaches that incorporate periodic mental health check-ins, motivational interviewing, and readily accessible counseling for employees with prior TWTD episodes. By aligning healthcare services, social security measures, and workplace policies, such integrated care pathways might expedite recovery and minimize relapses [37].

### 4.1. Strengths and Limitations

A principal strength of this study is its longitudinal design, enabling a detailed examination of TWTD trends and economic outcomes across distinct phases of the pandemic. The integration of cost analyses enhances the discussion by illustrating how mental health-related absences can reverberate through national and regional economies.

However, several limitations must be acknowledged: First, the retrospective nature of some data may introduce biases due to missing or incomplete records. The sample, derived from Asepeyo-affiliated workers, may not fully represent informal workers or those without an official mental health diagnosis, limiting generalizability. Additionally, regional disparities in healthcare access could have led to underdiagnosis and reporting variations, potentially skewing TWTD incidence rates. This must be recognized.

Economic fluctuations, evolving unemployment rates, and unmeasured psychosocial factors could have influenced TWTD incidence, complicating attempts to attribute changes solely to COVID-19 or workplace initiatives [38]. Lastly, focusing on clinically diagnosed disorders potentially overlooks mild or subclinical mental health issues that do not meet the formal criteria but still impact workplace performance. To address these limitations, future studies should do the following:Expand data sources to include other mutual insurance entities and national registries for broader workforce representation;Incorporate psychosocial risk factors such as job-related stress and coping mechanisms to assess their role in TWTD;Conduct longitudinal multicenter studies to compare trends across different occupational and healthcare settings;Evaluate sector-specific risks and gender disparities, particularly in high-risk occupations;Examine the long-term impact of workplace mental health interventions on TWTD incidence and relapse rates.

Future research can address these gaps, providing a more comprehensive understanding of TWTD and contributing to effective mental health policies in the workplace.

### 4.2. Implications for Practice

The observed decline in TWTD underscores the effectiveness of coordinated workplace mental health programs, flexible scheduling, and early intervention strategies in reducing absenteeism and its associated costs. Persistent gender disparities reinforce the need for tailored solutions that lessen the disproportionate emotional and logistical burdens on female workers, such as on-site childcare or job-sharing options. Targeted interventions—from psychoeducational sessions to return-to-work frameworks—can be particularly beneficial in high-stress environments like retail, care services, and healthcare.

Additionally, recent research highlights how physical activity, quality of life, and adequate sleep can ameliorate mental health challenges triggered or exacerbated by the COVID-19 pandemic [39]. Incorporating regular screening and low-threshold access to therapy or counseling within occupational health units might further curtail relapse rates, especially for GAD and major depression, and also strengthen overall workforce resilience. A tighter integration of public health, social security, and employer-based programs could yield a more robust safety net, supporting employees who face mental health challenges during crises and beyond.

## 5. Conclusions

The surge in mental health-related temporary work disabilities (TWTDs) during Spain’s strict COVID-19 lockdown, followed by a downturn in 2021–2022, highlights both the acute vulnerability of the workforce to pandemic stressors and the potential for gradual psychosocial recovery. However, persistent disparities—by region, gender, and occupational sector—indicate that pandemic-related mental health repercussions have not dissipated uniformly: these findings point to the need for targeted resource allocation, gender-sensitive approaches, and proactive mental health interventions to ensure equitable recovery across all workforce segments. This study provides valuable insights for occupational health policies by identifying critical risk factors for TWTD relapse and highlighting regional disparities in economic burden. From a theoretical standpoint, our findings contribute to the growing body of research on pandemic-induced mental health challenges, offering a nuanced understanding of how large-scale crises influence disability trends. Strengthening collaboration among healthcare providers, Mutual Collaborators with Social Security, and employers is essential to facilitate the early detection and efficient management of work-related mental disorders. Implementing flexible work policies, equitable pay structures, and evidence-based psychosocial support can help foster a more resilient workforce, ultimately reducing the economic impact of TWTD and promoting sustainable occupational well-being.

Moving forward, long-term follow-up studies will be crucial to determine whether TWTD rates remain stable, continue to decline, or rise again in response to emerging stressors. Cross-national comparisons could provide valuable insights into how different healthcare infrastructures and labor policies influence mental health outcomes during public health crises. Additionally, evaluating the effectiveness of targeted interventions, such as digital mental health platforms, structured return-to-work programs, and tailored psychosocial support for high-risk sectors, can generate evidence-based recommendations for employers and policy-makers. Finally, employing mixed-method designs that integrate qualitative perspectives on workplace culture, family–work conflict, and individual coping mechanisms may offer a deeper understanding of optimizing occupational mental health strategies both in crisis contexts and routine practice.

## Figures and Tables

**Figure 1 healthcare-13-00618-f001:**
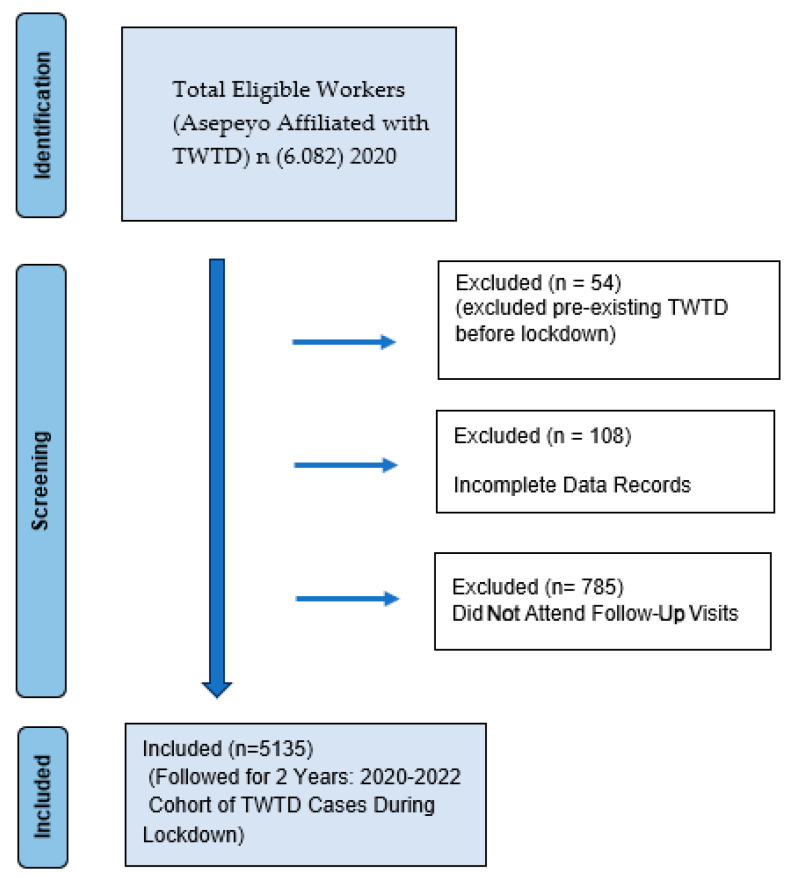
Flowchart of sample selection process.

**Figure 2 healthcare-13-00618-f002:**
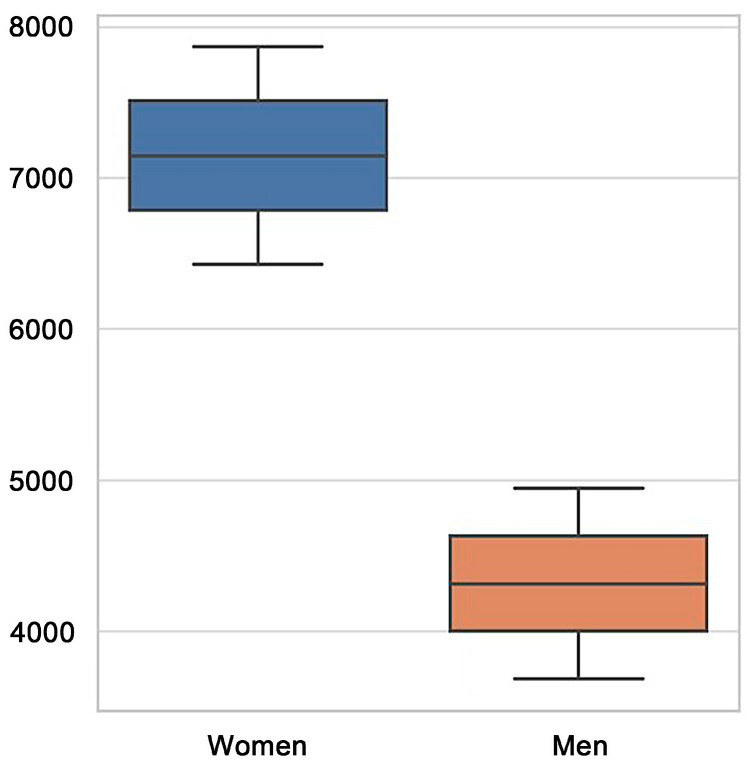
Total workdays lost due to TWTD (2021–2022) by gender.

**Figure 3 healthcare-13-00618-f003:**
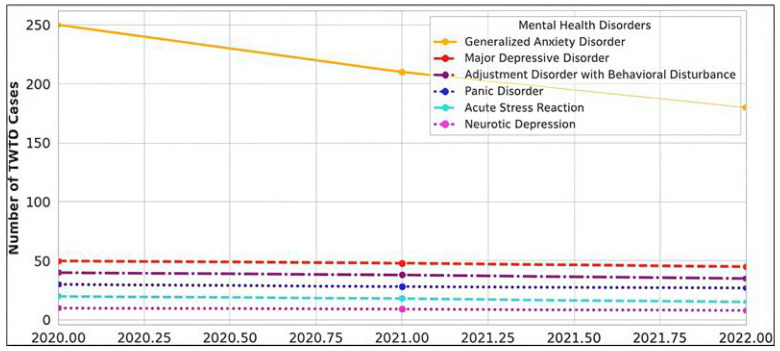
Evolution of the most prevalent mental health diagnoses (2020–2022).

**Figure 4 healthcare-13-00618-f004:**
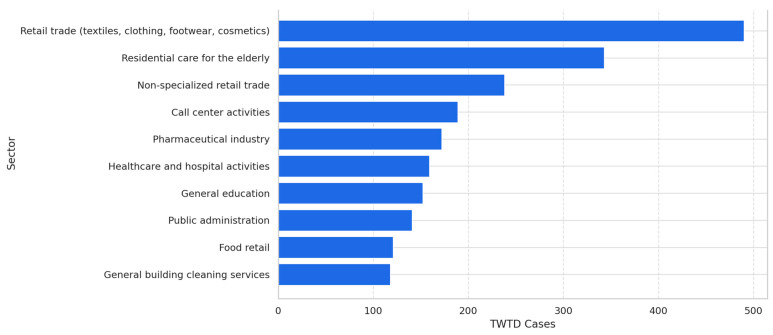
Top 10 sectors affected by TWTD cases in 2020.

**Table 1 healthcare-13-00618-t001:** Total economic burden of TWTD by Autonomous Communities (ACs), 2020.

Autonomous Community	Total Cost (EUR)	Percentage (%)
Andalusia	4,554,064.59	10.2
Aragon	1,000,818.58	2.2
Asturias	300,850.79	0.7
Canary Islands	380,603.58	0.8
Cantabria	703,018.35	1.6
Castile and León	4,847,655.33	10.8
Castile–La Mancha	6,781,918.3	15.1
Catalonia	14,466,062.46	32.3
Valencian Community	4,674,155.62	10.4
Basque Country	3,164,323.98	7.1
Extremadura	755,726.36	1.7
Galicia	484,107.11	1.1
Balearic Islands	227,322.2	0.5
La Rioja	397,706.66	0.9
Madrid	707,858.8	1.6
Murcia	442,915.22	1.0
Navarre	950,520.1	2.1

**Table 2 healthcare-13-00618-t002:** Average economic cost of TWTD per worker by gender and Autonomous Community (AC).

ACs	Total (EUR)	Women (EUR)	Men (EUR)	*p*-Value
Andalusia	1788.1	1606.8	2125.6	0.004 *
Aragon	2472.24	2157.6	3874.0	0.004 *
Asturias	2332.18	2040.3	2808.7	0.049 *
Canary Islands	2378.77	2002.1	2990.0	0.002 *
Cantabria	3390.5	2605.4	4736.3	0.016 *
Castile and León	2101.39	2098.4	2108.5	0.949
Castile–La Mancha	2180.23	2127.2	2298.0	0.388
Catalonia	2472.95	2305.2	2722.6	<0.001 **
Valencia	2268.53	2254.3	2289.6	0.875
Basque Country	2730.0	2189.8	3552.2	<0.001 **
Extremadura	1538.43	1681.7	1140.6	0.419
Galicia	2338.68	2308.2	2391.3	0.762
Balearic Islands	2621.75	2457.2	2972.9	0.326
La Rioja	1751.67	1385.8	2178.5	0.072
Madrid	2986.75	2761.4	3375.2	0.083
Murcia	2380.94	2254.7	2513.5	0.908
Navarre	2314.32	1952.8	2842.1	0.074

Data are shown as Euros. Significance is shown as ** *p* < 0.001 and significance *p* < 0.05 *.

**Table 3 healthcare-13-00618-t003:** TWTD cases, workdays lost, and economic cost (2021–2022).

Year	Total	Men (%)/Mean Age	Women (%)/Mean Age	COVID-19-Related TWTD (Men, %)	COVID-19-Related TWTD (Women, %)	Total Workdays Lost (Men/Women)
2021	308	40/45.3	60/43.6	2	6	4945/7866
2022	260	38/42.03	62/39.3	2	1	3685/6425

**Table 4 healthcare-13-00618-t004:** Economic costs related to TWTD (2021–2022).

Year	Total Cost (EUR)	Cost for Men (EUR)	Cost for Women (EUR)
2021	13,373,075.35	5,611,882.96	7,761,192.39
2022	1,089,587.52	621,598.08	467,989.44

## Data Availability

Aggregated data supporting this study’s findings are available upon reasonable request from the corresponding author, A.S.S., subject to review. These data are not publicly available due to privacy concerns and the potential for compromising research participant privacy/consent.

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
