# Peer review of "The Economic and Occupational Impact of Mental Health-Related Temporary Work Disabilities in Spanish Workers During and After the COVID-19 Pandemic: A Longitudinal Study"

_healthcare, 2025, doi:10.3390/healthcare13060618_

Round 1
Reviewer 1 Report
Comments and Suggestions for Authors
I have reviewed the manuscript entitled “The Economic and Occupational Impact of Mental Health-Related Temporary Work Disabilities During and After COVID- 19: A Longitudinal Study”. I have observed commendable work, but numerous comments remain for authors to consider enhancing the quality of this study.
- The methods section in the abstract is not clear; what was the data collection instrument? Also, what do you mean by CHAMAN? For the first time, it is recommended that you expand your abbreviation.
- The gap in the interaction is clear. However, the problem statement is not clear. How did you come out that temporary work disabilities were increasing at that time? Authors should support their claim with reliable statistics or recent studies that raised this issue, which existed in a spin.
- In method selection, don’t expect that all researchers know “longitudinal prospective design ”, so it is better to start with a definition of it.
- What was the volume of your population? And how did you identify the sampling size? What was your sampling size? I advise you to draw a flow chart for your method to make the phases of your data collection clear for future readers.
- In longitudinal studies, one of the essential tests is the paired T-test or ANOVA to determine the differences between groups or subjects throughout the period of your study. Can you justify why you did not use it?
- The limitations of this study are not clearly stated. Authors should write about the weaknesses of this study ended with suggestions for future research.
- The conclusion is okay. However, the practical and theoretical contributions of this study are missing
Comments on the Quality of English Language
It just need simple proofreading by a language expert
Author Response
Dear Reviewer 1,
We appreciate your feedback. Please find our attached response to your comment.
Best regards,

Reviewer 2 Report
Comments and Suggestions for Authors
GENERAL COMMENT
The Authors evaluated the economic and occupational impact of of mental health-related temporary work disabilities during the COVID-19 period. The study covers an interesting topic, the rationale is well established and the procedures are described in detail. However, there are some points that should be addressed to improve the manuscript’s fluency.
INTRODUCTION
LINE 58-71: please consider to expand the role of mental health impact due to COVID-19.
METHODS
LINE 137: please if it is possible consider to add some effect size measure.
DISCUSSION
Line 320: please consider to add a paragraph regarding the practical applications derived from main findings as the contribution of physical activity and health-related domains (quality of life, sleep) to ameliorate the workers’ mental-health status due to COVID-19 pandemic (e.g., Natilli M, Rossi A, Trecroci A, Cavaggioni L, Merati G, Formenti D. The long-tail effect of the COVID-19 lockdown on Italians' quality of life, sleep and physical activity. Sci Data. 2022 May 31;9(1):250)
Author Response
Dear Reviewer 2,
We appreciate your feedback. Please find our attached response to your comment.
Best regards,

Reviewer 3 Report
Comments and Suggestions for Authors
Dear authors, thank you for the opportunity to get acquainted with the results of your analytical study.
The article is devoted to the current topic of studying temporary disability related to mental health during and after COVID-19, as well as its economic consequences. The authors conducted an analytical study based on official statistics for the period from 2020 to 2022 in Spain.
1. Due to the fact that the study was conducted in Spain, it is important to reflect this in the title. It is also necessary to indicate in the title what category of employees we are talking about? And also explain what the authors put into the "professional effect"? The economic effect is clearly visible from the data in the article.
2. In the introduction, is it possible to add information about whether similar studies have been conducted in other countries? Perhaps not for such a period and not for all the parameters of comparison. But I would like to clarify this in order to understand the novelty of the results and their discussion.
3. I would like the authors to clarify why a number of figures and tables provide information only for 1 year, for example, Fig. 3. Because data for other years allowed us to identify the economic effect for specific sectors of the economy and professional groups.
The discussion of the results was conducted qualitatively, possible consequences were analyzed and measures were proposed, which makes the results valuable for practice.
The authors described the limitations of the study.
The conclusions correspond to the results obtained.
Best wishes, reviewer
Author Response
Dear Reviewer 3,
We appreciate your feedback. Please find our attached response to your comment.
Best regards,

Round 2
Reviewer 1 Report
Comments and Suggestions for Authors
I've noticed the authors have made significant improvements, which has greatly enhanced the overall quality of the manuscript. However, I was surprised that my comments about the methodology section, which is such a fundamental part of any scientific research, were overlooked. I kindly ask that you consider comment number 4 carefully.
"Comments 4: What was the volume of your population? And how did you identify the
sampling size? What was your sampling size? I advise you to draw a flow chart for your
method to make the phases of your data collection clear for future readers. "
Author Response
We appreciate your insightful comments.
Population Volume and Sampling Size Identification:
The total study population consisted of 6,082 Asepeyo-affiliated workers who experienced temporary work disability (TWTD) due to mental health conditions during the study period.
We applied strict inclusion and exclusion criteria to ensure data consistency and relevance.
Excluded cases included workers with pre-existing mental health-related TWTD before the pandemic (n=54), incomplete records (n=108), and those who did not attend follow-up visits (n=785).
After applying these criteria, the final sample consisted of 5,135 workers, who were followed for two years (2020–2022).
Flowchart for Clarity:
In response to your suggestion, we have included a flowchart in the manuscript to clearly illustrate the data collection process and sampling criteria, making it easier for future readers to follow the methodology.
We appreciate your feedback and believe these additions enhance the transparency and clarity of our study.

Reviewer 2 Report
Comments and Suggestions for Authors
The Authors addressed all suggestions properly. I don't have further comments.
Author Response
Dear Editor,
This reviewer explain: The Authors addressed all suggestions properly. I don't have further comments.